# Three-Dimensional Characterization of Polyurethane Foams Based on Biopolyols

**DOI:** 10.3390/ma16052118

**Published:** 2023-03-06

**Authors:** Lorenleyn De la Hoz Alford, Camila Gomes Peçanha de Souza, Sidnei Paciornik, José Roberto M. d’Almeida, Brenno Santos Leite, Harold C. Avila, Fabien Léonard, Giovanni Bruno

**Affiliations:** 1Department of Chemical and Materials Engineering, Pontifical Catholic University of Rio de Janeiro (PUC-Rio), Rio de Janeiro 22451-900, RJ, Brazil; 2Department of Basic and Biomedical Sciences, Simón Bolívar University, Barranquilla 080020, Atlántico, Colombia; 3Institute of Exact and Technological Sciences, Federal University of Viçosa, Viçosa 36570-900, MG, Brazil; 4Physics Program, Universidad del Atlántico, Puerto Colombia 081008, Atlántico, Colombia; 5Harwell Science and Innovation Campus, The University of Manchester at Harwell, Didcot OX11 0DE, Oxfordshire, UK; 6Bundesanstalt für Materialforschung und–prüfung (BAM), Unter den Eichen 87, 12205 Berlin, Germany; 7Institute of Physics and Astronomy, University of Potsdam, Karl-Liebknecht Str. 24-25, 14476 Potsdam, Germany

**Keywords:** biopolyol, banana, 3D microstructure, X-ray microtomography, compression mechanical

## Abstract

Two biopolyol-based foams derived from banana leaves (BL) or stems (BS) were produced, and their compression mechanical behavior and 3D microstructure were characterized. Traditional compression and in situ tests were performed during 3D image acquisition using X-ray microtomography. A methodology of image acquisition, processing, and analysis was developed to discriminate the foam cells and measure their numbers, volumes, and shapes along with the compression steps. The two foams had similar compression behaviors, but the average cell volume was five times larger for the BS foam than the BL foam. It was also shown that the number of cells increased with increasing compression while the average cell volume decreased. Cell shapes were elongated and did not change with compression. A possible explanation for these characteristics was proposed based on the possibility of cell collapse. The developed methodology will facilitate a broader study of biopolyol-based foams intending to verify the possibility of using these foams as green alternatives to the typical petrol-based foams.

## 1. Introduction

In recent decades, polyurethane (PU) has become a widely used material [1]. However, most PUs are derived from non-renewable resources and are detrimental to the environment. Polyols are an important raw material in this industry, and sustainable solutions are being sought that include their synthesis and characterization from a wide variety of vegetable oils [2,3,4,5].

Polyurethane foams (PUFs) are the most common example of polymeric foams. This type of polymer was synthesized circa 1849 when Charles A. Wurtz announced the synthesis of a substance that he called urethane at first that was the result of a reaction between an isocyanate group and a chemical that contained the hydroxy group (-OH) [6,7,8,9,10,11]. In 1937, the chemist Otto Bayer discovered polyaddition, studied these compounds, and produced polyurethanes [6].

The commercial revolution of PUFs began in Leverkusen (Germany) around 1930 with polyesters as polyols, leading to the development of rigid foams, adhesives, and paints. Flexible foams were developed in the 1950s, and around the 1970s, semi-flexible foams were in use (specifically in automotive production). In the 1990s, environmental concern grew over the amount of polluting material produced by these foams, and alternatives were devised to optimize their manufacture with recovery and reuse techniques [6,11,12,13,14].

Such foams are used predominantly due to their versatility and innumerable advantages in the aerospace (sandwich structures), building (construction materials, support, and thermal and acoustic insulation), and automotive industries (seats); they are also commonly used in packaging, electronic devices, furniture items, footwear, toys, and mattresses. In the case of sandwich-type structures or compounds, they typically consist of a three-layer arrangement with two sheets corresponding to the faces separated by a core generally made up of this class of foams. Due to their low density, they are characterized by their good thermal insulation and high durability even in chemically aggressive environments [6,7,12,15].

These foams generally comprise a polyol with di-isocyanate, distilled water, a surfactant element, and a catalyst (Figure 1). Due to environmental challenges, the amount of research based on sustainable polyols such as vegetable and biomass waste has increased recently. Biomass residues and byproducts from several industrial sectors can be a solution for the production of biopolyols, mainly because this solution is low-cost and there is no competition with food resources [16,17]

Gibson and Ashby addressed the micromechanical behavior of artificial and natural foams and showed that the physical properties of foams are directly linked to their microstructure. They introduced the concept of cellular materials, which are an assembly of cells with solid edges or faces that form a complex 3D microstructure—a foam [18,19].

One widely known technique for providing three-dimensional data on the internal structure of cellular solids is X-ray microtomography (microCT). MicroCT is non-destructive, requires little or no specimen preparation, and allows in situ experiments such as thermomechanical interaction with the sample. In this technique, the analyzed object rotates 360° around its axis, and a series of 2D X-ray absorption images are recorded. Using mathematical principles of tomography, the series of images is reconstructed and produces a 3D reconstruction (the digital images shown later in the text) of the sample [20]. 

Elliott, J.A. et al. (2002) [21] performed in situ deformation of polyurethane foam during a microCT scan at various compressive strains. They could also observe the behavior of the cells under compression. One of the purposes of the study was to construct a finite element model; however, it was not possible due to the lack of information about the sample’s boundary conditions, which influenced the elastic properties [21].

In 2004, E. Verhulp, B. Van Rietbergen, and R. Huiskes used CT images to study an open-cell aluminum foam. The method of tetrahedronization used allowed them to create a solid structure with tetrahedrons, and the displacements were calculated just in terms of each tetrahedron and converted into a deformation tensor at the center of each tetrahedron. A least-squares adjustment was carried out, which made it possible to determine and visualize the local strains in the deformations in the fabric images [14].

In 2007, Adrien, J. and Maire, E. studied the compression behavior of syntactic foams (including PU) via in situ microCT. Such foams are composed of glass microspheres embedded in a polymeric matrix. The experiments were performed at a high resolution using synchrotron radiation. 

Patterson, B.M. et al. (2016) [22] studied the mechanical properties of polymeric foams using in situ compression performed during synchrotron microCT. The experiment was conducted using hyperelastic polymers due to the importance of the 3D characterization of this type of structure and because the understanding of the dynamic response and mechanical stress is fundamental to predicting lifetime performance and stress recovery. The authors developed a technique to measure the morphological changes in their materials [22]. 

Even though these analytical techniques are well established in the analysis of foams in general, there is little research on the microstructure and properties of PU foams synthesized from polyols of vegetal origin. Among those properties, the mechanical response is of paramount relevance. Given the intrinsic variability of materials of vegetal origin and the complex 3D cell structure, microCT experiments with in situ mechanical tests are critical for establishing the link between the microstructure and the mechanical response [22].

This paper deals with novel sustainable biopolyols, provides new materials, and proposes a methodology for microstructural and mechanical characterization of this type of foam that employs microCT and in situ mechanical tests.

## 2. Materials and Methods

### 2.1. Materials

PU foams were produced from biopolyols obtained via liquefaction of samples of banana pseudostems (BS) and leaves (BL). The samples were dried at 105 °C in an oven until reaching a constant weight. Then, the biomass was cut in a knife mill to obtain 0.5 mm long pieces for future use [23]. The two components (leaf and stem) were fractions of the banana plant, and although they came from the same plant, they were from different locations and were therefore subjected to different environmental conditions, which is why they had specific morphologies and individual characteristics. Thus, these characteristics were transferred to the foams.

The crude glycerol used as a liquefaction solvent was provided by Petrobras (Usina Darcy Ribeiro, Montes Claros, MG, Brazil). Sulfuric acid (Synth) was used as the catalyst for this reaction. The reactants used to perform the hydroxyl number of the polyols produced were 1,4-dioxane (Synth), imidazole (Synth), phthalic anhydride (Synth), and sodium hydroxide (Synth). Polyurethane foams were synthesized via the batch process method using a mixture of biopolyol, isocyanate (Desmodur 44 V 20 of Bayer), surfactant (Tegostab 8460 supplied by Evonik, Essen, Germany), catalyst (Kosmos 19 commercialized by Evonik), and blowing agent (distilled water) with a mechanical stirrer (Fisatom model 713 D) until the complete homogenization of the system; the mixture remained under agitation for 10 s. The formulation was maintained in a mold for the growth of the polymer foam, and the mold was kept closed for 24 h at room temperature to cure [23]. 

When measuring the mass and volume, the density values obtained for the BS and BL samples were very similar: 0.17 ± 0.01 g/cm^3^ and 0.17 ± 0.03 g/cm^3^, respectively. The foams were cut into cylindrical shapes and subjected to traditional ex situ compression and in situ tests using X-ray microtomography. 

### 2.2. Ex Situ and In Situ Compression Tests

The initial ex situ tests were performed on samples of Group 1 by employing AME-2kN testing equipment with a capacity of 2 kN (200 kgf). The equipment was controlled by the DynaView Standard/ProM software (Version number V1. 05) and used an adaptation of two smooth plates (one fixed and the other mobile). The test followed the ASTM-D-1621 standard [24]. The samples were cylindrical with a diameter of 8 mm and a height of 16 mm. 

The in situ system was a universal mechanical testing device (CT5000, Deben, UK) designed for mechanical testing in microtomography systems (Figure 2a,b). The Microtest stage control software monitored this system. The specimen was fixed between two aluminum flanges; the lower flange moved vertically to compress the sample. The tomograph was a GE VTomeX operating at 70 kV, 140 µA, and a 16 µm pixel size. The samples were also cylindrical with a diameter of 20 mm and a height of 40 mm.

The experiments were performed in two stages with duplicate samples. First, the compression response curve of one sample of each kind was determined while acquiring a sequence of X-ray projection images at a fixed sample angular orientation. This acquisition mode required a relatively short exposure time (0.5 s) for each image, which allowed a continuous compression test at a constant speed of 0.1 mm/min. Even though this response curve was similar to the one obtained in the ex situ tests, it was necessary because it reflected the conditions of the specific stage used in the tomograph. Specific displacement points were selected to serve as a reference to the second stage. These points were (0; 0.5; 1; 2; 4) mm for BS8 and (0; 0.5; 1; 1.5; 3) mm for BL6. These displacements were converted to deformation values (in %) in the images and plots shown in the Results section.

Then, the second sample was placed in the in situ stage, which was programmed to stop at the chosen displacement points, and a full tomogram was acquired at each point. This required 2400 projections and a total scan time of 20 min at each point. 

### 2.3. Image Processing and Analysis

Image processing and analysis were performed using FIJI/ImageJ and ORS Dragonfly (Dragonfly 4.1, Object Research Systems, Montreal, QC, Canada) following the typical pre-processing, segmentation, post-processing, and quantification steps [24]. The main goal was to discriminate individual foam cells and then measure their size and shape parameters for the various in situ compression points. The experiment was divided into two stages. The first consisted of preliminary compression tests with samples to determine the mechanical response to compression. For the second stage, the subgroup of samples with the highest compression limits was tomographed and tested in situ (see above). 

Figure 3 shows some of these steps for a small cutout of one of the 2D layers. Pre-processing comprised noise reduction with the edge-preserving non-local means filter [25] (see Figure 3a,b). Noise reduction was relevant to eliminating spurious points leading to fake cell walls. Segmentation was carried out with a manually defined intensity threshold. Given the excellent contrast between cell walls and their interior spaces and the stability in brightness and contrast across all tomography layers, it was possible to use a fixed threshold. Figure 3c shows the result of segmentation.

Even though cell wall segmentation was robust, identifying and measuring individual cells was still challenging due to the difficulty in obtaining fully closed contours and thus separated cells. The arrows in Figure 3c highlight some discontinuities in cell walls. These discontinuities may have been real or artifacts due to limited resolution. In other words, adjacent cells might have merged, leading to an incorrect assessment of the cell size and shape. The classic solution to this kind of problem is watershed segmentation [26]. In the present case, a 3D algorithm available in Dragonfly was used. It used a 3D distance map and a manually defined threshold to create seeds for individual cells that grew without further merging [27,28]. 

Figure 4 shows a complete microCT layer of the BS sample after the entire image-processing sequence. Even though the watershed step also created false boundaries, this was an acceptable trade-off compared to the amount of cell merging without cell separation. Thus, the cell number, size, and shape measurements for different compression steps were performed on the watershed-separated 3D images.

The cell size was measured according to the 3D volume (V), and the cell shape was characterized according to the aspect ratio (AR). The AR was the ratio between the minimum and maximum Feret diameters (projections) and mainly described the cell directional isotropy, which varied from 0 for highly elongated objects to 1 for an isotropic object. It is essential to mention that, given the variety of cell shapes in 3D, the minimum and maximum Feret diameters could lie along any direction in space and did not need to be aligned with the stress direction or a plane orthogonal to it.

## 3. Results and Discussion

Figure 5 shows the stress–strain curves obtained in the ex situ compression tests. Three samples of each kind were measured. The overall response variation among similar samples was typical of polymers of lignocellulosic origin. The oscillations likely arose from sequential cell wall rupture or collapse during compression. Table 1 shows the average values for the Young’s modulus (E) and ultimate strength (LR) for each sample type. These numbers indicated a similar macroscopic compression response of foams produced from banana stems and leaves. However, the samples were quite different from a microscopic perspective as shown by the oscillatory behavior of the stress–strain curves of the BL samples compared to the smooth behavior of the BS ones. Such behavior was symptomatic of a collective cell collapse and was typical of cellular materials with large cells (i.e., a large ratio between the cell diameter and strut thickness). This point will be discussed below.

Table 2 shows some properties of different types of polyurethane synthesized using bio-based polyol and petroleum-based polyol. Tavares compared polyurethanes prepared by combining Kraft lignin and modified castor oil with different compositions; PU-MCO2/L30 (modified castor oil obtained by adding 30% by weight of technical Kraft lignin) had the highest elastic modulus and a large difference with the petroleum-based example. When comparing their results with those of this study, it was concluded that the range of mechanical properties (Young’s modulus) varied greatly with the chemistry and synthesis of the polyurethane depending on the type of polyol used [29].

### 3.1. In Situ Tests and Visualization

Figure 6 shows a radiographic sequence that revealed the deformation of the BS sample. These radiographs were captured during the determination of the response curve of the sample in the in situ holder. The bottom plate can be seen moving up during the experiment. The images reveal a few relevant characteristics. The dark bands correspond to foam cell damage as the compression increased. As this damage grew, the mechanical response of the sample changed, and the specimen began to shear and buckle, which deviated from purely uniaxial compression conditions.

Figure 7 shows the 3D tomographs of the BS sample for specific deformation values of the compression curve. The overall reduction in sample height and increase in the sample width is visible. This sample also showed some shear and buckling but to a lesser extent. A large pore and other smaller ones are visible at the surface. These defects most likely were formed during nucleation or thermal expansion of the polyurethane or in the final drying process. 

The 3D images highlight the complexity of the microstructure and justify the need for in situ mechanical tests. A reliable quantitative analysis that correlated with the compression conditions was only possible if performed continuously on a single sample. Any interruption, removal of the sample from the holder, or change in sample would prevent obtaining comparable measurements (data registration was extremely difficult due to the absence of ‘trackers’; i.e., load-invariant features). 

### 3.2. Quantitative Analysis 

Figure 8 shows representative 2D cross-sections of the 3D reconstructions for each sample after the complete image-processing sequence. The visualization of the watershed cell separation revealed a significant difference in cell size between the BS and BL samples. It should be noted that watershed cell separation is the most critical and can lead to super segmentation. New approaches such as the Deep Watershed may improve the method’s robustness [28]. 

This difference was reflected in the total number of cells, which was roughly 5X larger in the BL sample. Figure 9a,b show these differences and the evolution in the number of cells with increasing deformation for both samples. The number of cells increased with deformation (approximately 8.5% for BS and 20% for BL) when compressing from 0 to 10% and 7.5% mm, respectively. This was confirmed by the decrease in the mean cell volume (Figure 9c,d). The mean cell size decreased as the compression test evolved. For BS, the mean volume fell by 10.5% when the deformation increased from 0 to 10%. The average volume dropped by 20.3% between 0 and 7.5% deformation for BL.

This increase in the number of cells can be partially explained by the collapse of cell walls during compression (as illustrated in Figure 10). A cell may be broken into smaller ones, thereby increasing the number of individual cells. 

Figure 11 shows the shape distribution based on the 3D aspect ratio for each deformation step of the two samples. The 3D AR was the ratio between the smallest eigenvalue and the largest for the inertia eigenvectors of an object. The mean for each distribution is highlighted in the black boxes to make the comparison easier. Both samples tended to have anisotropic cells (3D AR < 0.5). The BS sample (3D AR~0.40–0.42) had slightly more elongated cells than the BL sample (3D AR~0.47–0.48). 

Interestingly, there was minimal shape variation with compression for both samples. As the load increases and the cells are compressed along the vertical axis, one would expect the minimum length to lie along the compression axis and the maximum axis on the orthogonal plane as the cells spread out. Thus, 3D AR should decrease as the load increases, but that did not seem to be the case. 

There are several possible explanations. First, as schematically shown in Figure 10, cell collapse can create smaller and fewer anisotropic cells. The 3D AR for the original cell (Figure 10a) was ≅0.5, while for the two cells in Figure 10c, it was slightly larger (≅0.7). Second, as mentioned before, the aspect ratio does not necessarily measure deformation parallel to the compression axis because it is the ratio between the minimum and maximum axes, and for a 3D shape, these can lie in any direction in space. Finally, as shown in Figure 6 and Figure 7, the samples showed some shear and buckling during compression, which likely affected the cell deformation.

A limitation of the proposed methodology derives from the typical trade-off between microCT resolution and image volume (the so-called field of view) because the resolution is inversely proportional to the sample thickness. A higher spatial resolution would provide better detail of the cell walls and improve cell separation but limit the analyzed sample volume, which can be critical for a non-uniform material. This was left for further work.

## 4. Conclusions

This paper focused on the compression response of foams produced from banana stems or leaves. Initial ex situ compression tests showed a similar macroscopic mechanical behavior for both kinds of samples. To better correlate the mechanical behavior to the microstructure, the samples were submitted to in situ compression tests during X-ray microtomography. The quantitative analysis of the 3D images required processing steps involving noise filtering, segmentation, and separation of touching cells. 

It was shown that BL-based foams had cells that were roughly 5× smaller than those of BS-based foams. The number of cells increased for both foams (by 8.5% and 20% for BS and BL, respectively), and the average cell volume decreased (by 10.5% and 20.3% for BS and BL, respectively) with increasing compression, while the cell aspect ratio remained essentially constant. A possible mechanism of cell collapse was proposed to explain part of this behavior. 

In addition, a methodology was developed to fully segment the tomography reconstructions without incurring artifacts such as super-segmentation. Such an approach allowed quantitative analysis of the cell geometrical features reported above and applies to similar systems (e.g., porous and granular materials).

To confirm the collapse mechanism proposed above, one would need higher-resolution 3D images showing details of the cell walls such as their thickness, formation of triple points, and other local characteristics. It is hard to specify the ideal resolution based on the experiments performed so far, but we believe that a resolution close to 2 µm would suffice.

To be reported elsewhere, new experiments will take advantage of the lens setup of a Zeiss Xradia 510 Versa tomograph (Carl Zeiss, Pleasanton, CA, USA), which permits a higher resolution (up to ≅1 µm) and volume stitching by partially bypassing the standard geometrical setup limitation.

In addition, new experiments using several other foams produced with biopolyols from various vegetal sources are underway that include the methodology outlined above and thermal and acoustic characterization. The goal is to identify the best biopolyol-based foams to replace traditional foams in various industrial fields.

## Figures and Tables

**Figure 1 materials-16-02118-f001:**
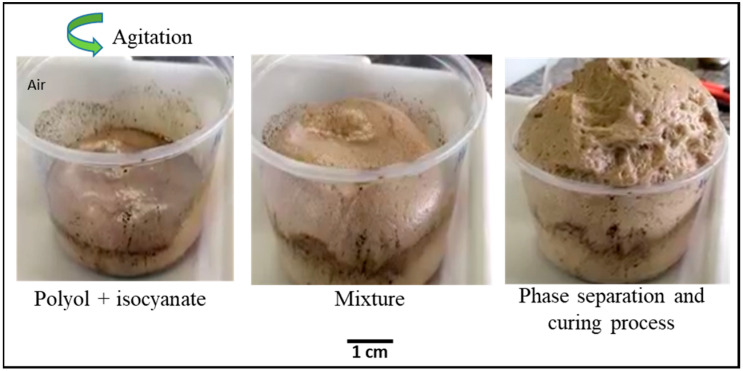
On the left the step by step of the PU foam production process.

**Figure 2 materials-16-02118-f002:**
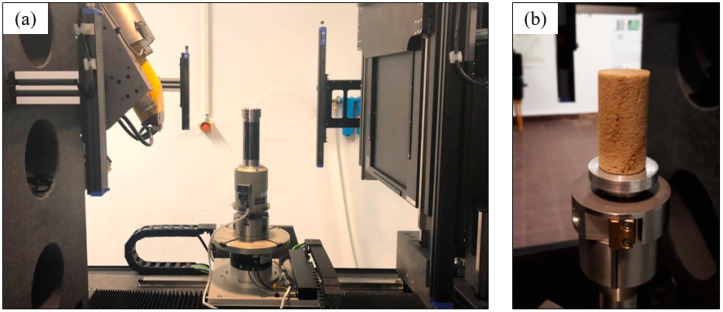
MicroCT in situ compression setup. (**a**) The in situ mechanical stage is shown between the X-ray source and the detector. (**b**) Close-up view of the foam sample positioned in the stage base.

**Figure 3 materials-16-02118-f003:**
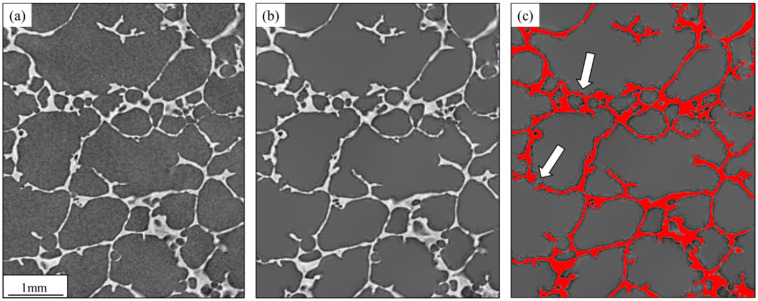
Small cutout of a 2D tomography layer to show the effect of noise reduction and segmentation: (**a**) original image; (**b**) after the application of the non-local means filter. Notice the excellent edge preservation. (**c**) Segmentation of cell walls (highlighted in red). The arrows point to incomplete/broken walls.

**Figure 4 materials-16-02118-f004:**
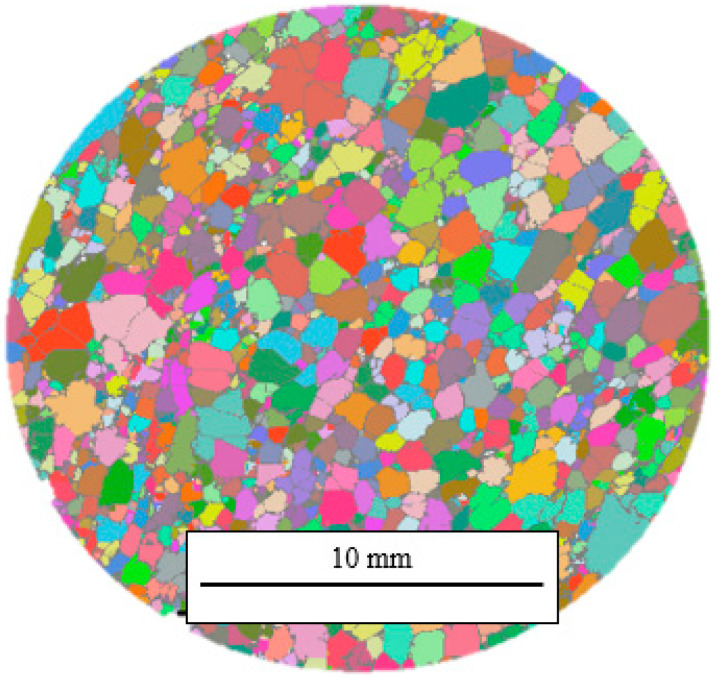
A complete microCT layer of the BS sample after 3D watershed segmentation. Individual foam cells are shown in different colors. The software automatically picked the colors to help distinguish between adjacent cells.

**Figure 5 materials-16-02118-f005:**
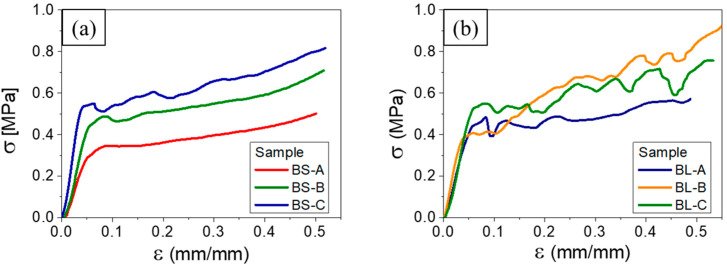
Results of the ex situ compression tests: (**a**) BS samples; (**b**) BL samples.

**Figure 6 materials-16-02118-f006:**
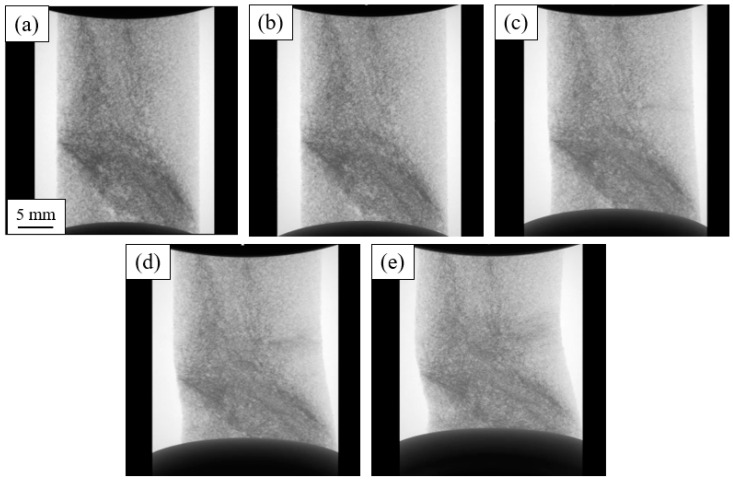
BS sample. Radiographic evolution of the deformation for each deformation: (**a**) 0%; (**b**) 1.25%; (**c**) 2.5%; (**d**) 5%; (**e**) 10%.

**Figure 7 materials-16-02118-f007:**
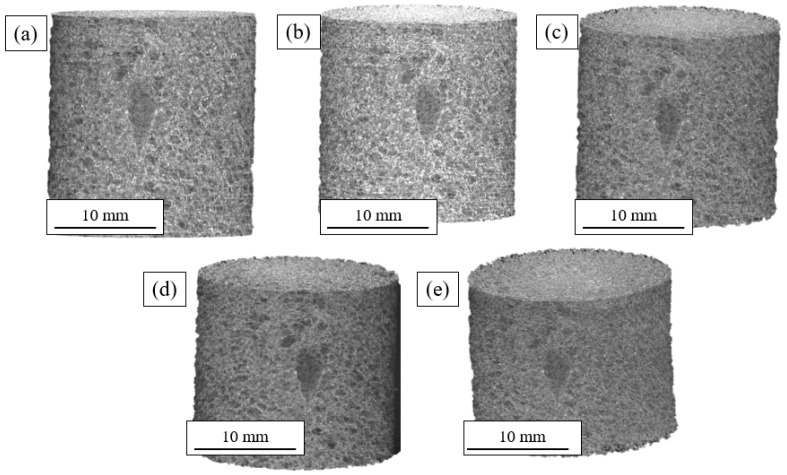
BS sample. Three-dimensional reconstructions of the evolution of the deformation for each deformation: (**a**) 0%; (**b**) 1.25%; (**c**) 2.5%; (**d**) 5%; (**e**) 10%.

**Figure 8 materials-16-02118-f008:**
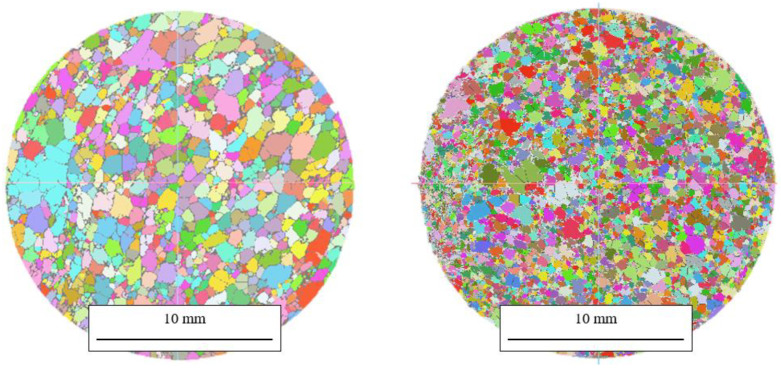
Typical 2D cross-sections of the 3D reconstructions for the BS (**left** image) and BL samples (**right** image). The smaller cell size for BL is visible. The software automatically picked the colors to help distinguish between adjacent cells.

**Figure 9 materials-16-02118-f009:**
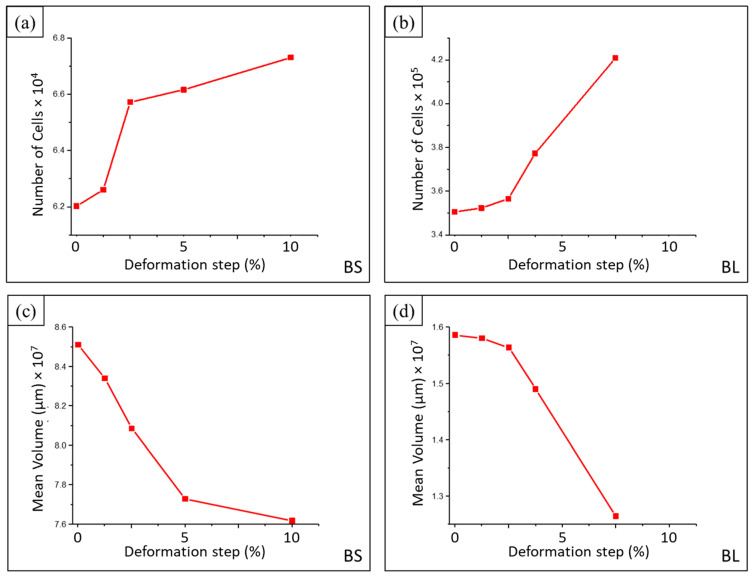
Evolution of the number of cells for each deformation step for (**a**) BS and (**b**) BL and of the mean cell volume for (**c**) BS and (**d**) BL.

**Figure 10 materials-16-02118-f010:**
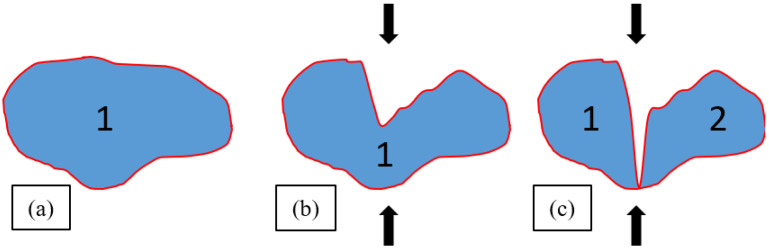
Illustration of cell collapse due to compression: (**a**) original cell shape; (**b**) cell deformation; and (**c**) cell collapse, which leads to an increase in the number of individual cells (The number 1 represents the studied cell and 2 the cell that is generated after the collapse).

**Figure 11 materials-16-02118-f011:**
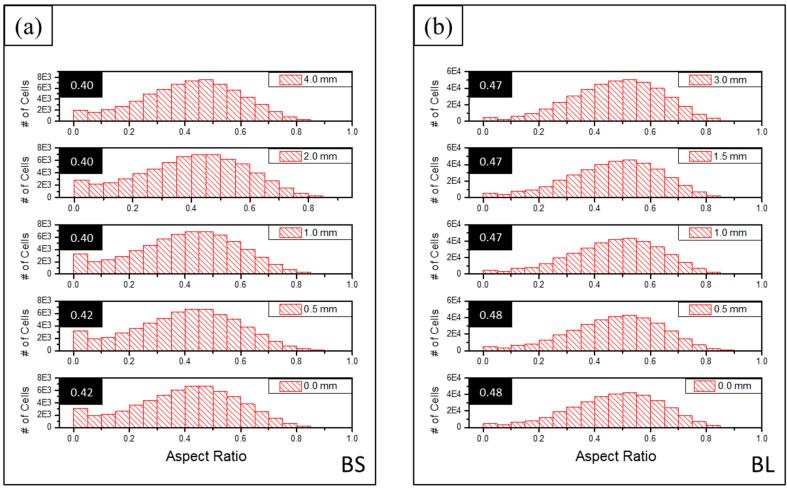
Cell 3D aspect ratio distribution as a function of the deformation step: (**a**) BS; (**b**) BL. The numbers in the black boxes are the mean values for each distribution.

**Table 1 materials-16-02118-t001:** Average values of $\bar{E}$ and ${\bar{L}}_{R}$ resulting from the stress–strain curves of the ex situ compression tests.

Sample	E (Mpa)	L_R_(Mpa)
**BS**	11.22 ± 3.54	0.67 ± 0.16
**BL**	10.72 ± 1.06	0.76 ± 0.20

**Table 2 materials-16-02118-t002:** Young’s moduli of polyurethanes based on different types of polyols.

Polyurethane Type	E (MPA)
**PU-BS**	10.72
**PU-BL**	11.22
**Polyurethane (Petrochemical Polyol)**	0.73–2.25
**PU-MCO2/L30**	2000

## Data Availability

Not applicable.

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
