# Peer review of "Three-Dimensional Characterization of Polyurethane Foams Based on Biopolyols"

_materials, 2023, doi:10.3390/ma16052118_

Round 1

Author Response

Dear reviewer,

First, we would like to thank you for allowing us more time to submit the revised manuscript. Below you will find the response to the comments and observations you made.
We did our best to respond and accommodate the request.
Naturally, we will be happy to provide you with further clarification, if you deem it necessary.

Sincerely,

Lorenleyn De la Hoz Alford, Ph.D.
Professor at the Simón Bolívar University (Barranquilla, Colombia)

Author Response

(The authors gave the same response as above.)

Reviewer 3 Report

Combining two keywords, the use of eco-friendly materials and 3D image analysis, can be a very good idea. However, there are a few things to consider.

A. It is analyzed without an accurate definition of the material.

A-1. Are the A, B, and C conditions used in the experiment the same sample?

A-2. BS and BL show different tendencies. What is the material difference?

A-3. If you want to explain the characteristics of biopolyols, shouldn't the comparison group be accurately represented?

A-4. In conclusion, we do not know why BS and BL show different trends, and what differences they have from common polyols.

B. Overall, the explanation of the experimental results is insufficient.

B-1. In Figure 6, where specifically does shear break occur?

B-2. Is Figure 10 really necessary?

B-3. It is expected that more diverse data can be obtained if the 3D analysis is performed, but aren't the results showing too definite?

B-4. Is it significant that the average value of the Aspect Ratio is maintained? If it is a mono-cell, there will be a characteristic for deformation, but it is judged that it will not be of great significance in a multi-cell. It seems to be used as a basis for cell collapse.

C. Additional comments

C-1. Wouldn't it be better to convert the Deformation Step to % rather than mm?

C-2. There are no specific material requirements for preparing PU.

Author Response

(The authors gave the same response as above.)
